# Potential of Extracellular Vesicle-Associated TSG-6 from Adipose Mesenchymal Stromal Cells in Traumatic Brain Injury

**DOI:** 10.3390/ijms21186761

**Published:** 2020-09-15

**Authors:** Santiago Roura, Marta Monguió-Tortajada, Micaela Munizaga-Larroudé, Marta Clos-Sansalvador, Marcella Franquesa, Anna Rosell, Francesc E. Borràs

**Affiliations:** 1ICREC Research Program, Health Science Research Institute Germans Trias i Pujol, Can Ruti Campus, 08916 Badalona, Spain; mmonguio@igtp.cat (M.M.-T.); mmunizaga@igtp.cat (M.M.-L.); 2CIBERCV, Instituto de Salud Carlos III, 28029 Madrid, Spain; 3Department of Medicine, Universitat Autònoma de Barcelona (UAB), 08193 Barcelona, Spain; 4REMAR-IVECAT Group, Health Science Research Institute Germans Trias i Pujol, Can Ruti Campus, 08916 Badalona, Spain; mclos@igtp.cat (M.C.-S.); mfranquesa@igtp.cat (M.F.); 5Department of Cell Biology, Physiology and Immunology, Universitat Autònoma de Barcelona (UAB), 08193 Cerdanyola del Vallès, Spain; 6Nephrology Service, Germans Trias i Pujol University Hospital, 08916 Badalona, Spain; 7Neurovascular Research Laboratory, Vall d’Hebron Institut de Recerca, Universitat Autònoma de Barcelona (UAB), 08193 Cerdanyola del Vallès, Spain; anna.rosell@vhir.org

**Keywords:** conditioned medium, extracellular vesicles, immunomodulation, mesenchymal stromal cells, traumatic brain injury, paracrine function, regenerative medicine, tissue repair, translational medicine, tumor necrosis factor-stimulated gene-6

## Abstract

Multipotent mesenchymal stromal cells (MSC) represent a promising strategy for a variety of medical applications. Although only a limited number of MSC engraft and survive after in vivo cellular infusion, MSC have shown beneficial effects on immunomodulation and tissue repair. This indicates that the contribution of MSC exists in paracrine signaling, rather than a cell-contact effect of MSC. In this review, we focus on current knowledge about tumor necrosis factor (TNF)-stimulated gene-6 (TSG-6) and mechanisms based on extracellular vesicles (EV) that govern long-lasting immunosuppressive and regenerative activity of MSC. In this context, in particular, we discuss the very robust set of findings by Jha and colleagues, and the opportunity to potentially extend their research focus on EV isolated in concentrated conditioned media (CCM) from adipose tissue derived MSC (ASC). Particularly, the authors showed that ASC-CCM mitigated visual deficits after mild traumatic brain injury in mice. TSG-6 knockdown ASC were, then, used to generate TSG-6-depleted CCM that were not able to replicate the alleviation of abnormalities in injured animals. In light of the presented results, we envision that the infusion of much distilled ASC-CCM could enhance the alleviation of visual abnormalities. In terms of EV research, the advantages of using size-exclusion chromatography are also highlighted because of the enrichment of purer and well-defined EV preparations. Taken together, this could further delineate and boost the benefit of using MSC-based regenerative therapies in the context of forthcoming clinical research testing in diseases that disrupt immune system homeostasis.

## 1. Introduction

Adverse immune responses broadly impair damaged tissue regeneration and are a major drawback in human organ and tissue transplantation leading to organ failure and death [1]. Mostly, tissues following mechanical injury suffer from an intense inflammatory response induced by damage-associated molecular patterns released by dead and dying cells [2]. In general, this extremely complex process is required for the clearance of dead cells and debris but has to be timely suppressed to allow tissue regeneration and repair. In this context, a broad range of cell types orchestrate this “physiological” or “sterile” inflammation process, including neutrophils, macrophages, innate lymphoid cells, natural killer (NK) cells, B cells, T cells, fibroblasts, epithelial cells, endothelial cells, and reparative stem cells.

Mesenchymal stromal cells (MSC) constitute a heterogeneous cell population of non-hematopoietic progenitors, with a fibroblast-like appearance, plastic adherence capacity, ability to differentiate into tissues of mesodermal lineages, and a well-established cell surface expression pattern, regardless of the tissue from which they are collected [3]. MSC have been largely associated with both immunomodulatory and regenerative capabilities in vivo [4,5]. However, clinical benefit through application of MSC is far from being completely reached, partly due to the remarkable heterogeneity of MSC sources and the lack of specific biomarkers predicting the success of prospective MSC-based therapeutic products. Moreover, together with autocrine factors released by MSC in the microenvironment [6], evidence supports that the contribution of MSC is mainly driven by paracrine signaling, rather than a direct cell-to-cell effect. This assumption is consistent with the preclinical observation that a low number of retained cells promotes beneficial effects.

Here, we highlight the powerful potential of MSC for immune suppression and tissue regeneration through a number of mechanisms of action such as those driven by a plethora of membrane-bound and secreted molecules such as tumor necrosis factor (TNF)-stimulated protein 6 (also known as TNF-stimulated gene-6 or TSG-6) and those encapsulated in the nanosized, bilayered endogenous messengers referred to as extracellular vesicles (EV). In this context, we would like to add further potential to the original research that focused on the increase of MSC therapeutic potential by means of modulation of TSG-6 expression and use of highly purified nanosized EV. This is, for instance, the case of interest in treatment of mild traumatic brain injury (mTBI).

## 2. Traumatic Brain Injury: A Representative Potentially Mesenchymal Stem Cell-Treated Disease

TBI is a common event with an average incidence in Europe of 262 per 100,000 people which occurs when a sudden trauma damages the brain and disrupts normal brain function. Recently, it has been redefined as an alteration in brain function, or other evidence of brain pathology, caused by an external force [7]. TBI can be classified according to the Glasgow Coma Scale (GCS) score, which is based on level of consciousness [8], as severe (GCS < 9), moderate (GCS 9–12), or mild (GCS 13–15), with mild TBI being the more common (>70% of all cases) [9,10]. Patients with mild TBI usually show no abnormalities in their neural tissue when explored by common imaging techniques such as computed tomography or magnetic resonance imaging, being asymptomatic within days to weeks of the injury. However, many of them experience persistent post-contusive symptoms, including headache, fatigue, sleep disturbances, dizziness, as well as cognitive and behavioral difficulties [11].

In addition, TBI consists of the following two types of damage: The primary type is caused by the mechanical forces at the time of the trauma that provoke multiple harmful effects such as shearing, tearing, and stretching of neurons, axons, glia, and blood vessels; and the secondary or nonmechanical type is caused by both ischemic-related biochemical and cellular alterations [12] which include inflammation, apoptosis, and oxidative stress, leading to fluid retention or swelling, new hematomas, and increased intracranial pressure that expand the damage to surrounding areas. In this context, cerebral hypoxia, understood as a decrease of oxygen supply to the brain, can develop as a secondary complication over a period of hours to days after TBI, being more frequent in moderate or severe injuries. Remarkably, these hypoxia episodes can aggravate initial brain damage (with increased neuronal death and neuroinflammation), and correlate with worse neurological outcomes. Then, late occurrences post-injury trigger peripheral inflammatory and immunomodulatory cells to enter the brain and further increase inflammatory activation by a variety of released pro- and anti-inflammation molecules. Although it is less frequent in mild TBI, there can be a third phase of complications primarily associated with seizures and epileptogenesis which arises weeks or months following TBI. 

Importantly, the delayed nature of secondary injuries offers an opportunity for supportive care and a therapeutic window for treatments to decrease the impact of brain damage and improve patients’ outcomes. With regard to common treatments, two main mechanisms are mainly targeted: neuroinflammation and neurorepair; the first one to reduce the damage and acutely protect the brain tissue under risk, and the second one to boost endogenous restorative processes which are potentially activated to ensure brain tissue remodeling and recover of neurological functions [13]. This biphasic neurorestorative treatment scheme should be continuous and interconnected over time for a successful development of novel therapies against TBI. An example is an early approach to inhibit inflammation, which at high doses or in the long term leads to worse outcomes since inflammation could have both detrimental and beneficial effects [14]. In brief, the major pathological changes of the brain tissue after mTBI include resident microglia activation and impairment of the blood-brain barrier which allow the entry of circulating neutrophils, monocytes, and lymphocytes to the injured site and directly exacerbate neuronal death by the entrance and on-site release of various neurotoxic substances such as prostaglandins, free radicals, complement factors, and pro-inflammatory cytokines.

Over the last decade, cell-based therapies have received considerable attention to improve TBI outcome as they can modulate inflammation, stimulate endogenous repair mechanisms, and potentially differentiate into new functional cells [13]. It is well established that oxygen deprivation, creating a hypoxic environment, is crucial to influence the properties of MSC [15]. This is, for example, the case of the enhanced proliferation and angiogenic potential of MSC observed in hypoxia preconditioned MSC from adipose tissue. Additionally, under hypoxic conditions, MSC are induced to secrete more lactate, which polarizes macrophages to an anti-inflammatory phenotype, and exhibit great amounts of IFNy-mediated IDO and their capacity to inhibit T cell activation. Thus, in the context of TBI, the absence of oxygen could be responsible, in part, for the therapeutic benefit of MSC therapy, including the enhancement of their immunomodulatory capacity. As discussed in the following pages, MSC and their secreted derivatives termed EV could potentially enhance neurorestoration after TBI through its immunomodulatory and reparative actions on injured tissues at the same time, making them excellent candidates to be applied in clinical practice. 

## 3. Immunomodulatory Effects of Mesenchymal Stem Cells

Since their foundation as a scientific discipline with the discoveries of phagocytosis and neutralizing antibodies, the development of immunology has experienced an authentic revolution throughout the last century, and this has contributed greatly to important advances in biomedicine. In this context, the well-reported immunomodulatory properties of MSC make them very promising on the stage for the repair of tissue and organ damage caused by chronic inflammation or autoimmune disorders.

In general, the immunomodulatory effect of multipotent MSC, either preconditioned with inflammatory cytokines or apoptotic, metabolically inactivated or even fragmented cells, is transmitted via a variety of secreted cytokines or factors, and surface molecules (Table 1) [16,17,18,19,20,21,22,23,24,25,26,27,28,29,30,31,32,33,34,35,36,37,38,39,40,41,42,43,44,45,46,47,48,49,50]. As a consequence, MSC have the ability to interact and change the functional properties of diverse host immune cells, either from the innate or the adaptive immune system (Table 2) [51,52,53,54,55,56,57,58,59,60,61,62,63,64,65,66,67,68,69,70,71,72,73,74,75,76,77,78]. Concisely, compelling evidences have demonstrated that inhibition of neutrophil infiltration, oxidative burst and extracellular trap release [26,30,45,46,47], reduction of NK cell activation and cytotoxicity [31,41,44,79], and C3b-regulatory protein Factor H-mediated complement system activation [24,34] are included in the specific mechanisms by which MSC actively modulate the innate immune system. In addition, they regulate effector cellular immune responses by distinct signaling pathways, including inhibition of mitogenic, antigenic and allogeneic T cell proliferation, reduction of T cell migration and cytotoxic activity, and enhancement of genetically regulated T cell self-destruction [18,22,29,30,33,37,38]. Moreover, MSC modulate T cells plasticity from Th1 and Th17 towards Th2, and therefore generate regulatory T cells through the production of a variety of anti-inflammatory cytokines in a M2 monocyte-mediated process [24,30,39,41,44,58,61,62,63,64]. Specifically, the presence of MSC in a pro-inflammatory microenvironment actively induces the transdifferentiation of both monocytes and macrophages towards an anti-inflammatory phenotype, specifically termed M2 [24,52,80,81,82], as well as are capable of impairing both differentiation and maturation of dendritic cells (DC) [40]. Molecularly, MSC provoke the shift of the pro-inflammatory milieu induced by extracellular ATP to the anti-inflammatory regulation by purine nucleoside adenosine (Ado) through a specific CD39/CD73-mediated hydrolysis [83,84,85,86]. This is, for instance, the case after the local implantation of a tissue engineering MSC-based graft that promotes myocardial infiltration of host CD73-positive monocytes in swine, as reported by Monguió-Tortajada et al. [5]. By using an in vitro co-culture setting, the same authors confirmed a polarization phenomenon of monocytes differentiating towards a regulatory M2 phenotype via de novo expression of the ectonucleotidase CD73, which is also a cell marker for MSC linked to T and NK cells regulatory phenotypes [86,87,88]. 

Taken together, these immunomodulatory properties are essential to unquestionably identify MSC as potential reparative biologicals for application after tissue injury or to avoid unwanted graft rejection in organ transplantation in spite of their short lifespan upon in vivo administration. For instance, once injected intravenously, MSC do not migrate across the lung barrier and get trapped because of their large size, and the fact that they are rapidly eliminated by monocytes/macrophages [89,90,91]. This theoretically limits the long-lasting action of infused cells and could generate pulmonary thromboembolism. For that, potential thrombolytic or anticoagulant regimens are needed, in parallel, for safer MSC-based applications and to maximize clinical benefit for the patients. MSC are, however, able to promote paracrine immunosuppression and tissue repair through modulation of recipient immune cells by a number of secreted factors such as IL6, PGE2, TGFβ, IDO, HGF, HLA-G, and TSG6, as well as a variety of double-layer phospholipid membrane vesicles carrying a variety of proteins and RNA [90,92,93]. Specifically, Ado production is also part of the immunosuppressive activity of MSC reducing inflammation, due to the fact that Ado can be shed from the plasma membrane, acting in its soluble form or released inside paracrine vesicles [17,94,95,96,97]. Furthermore, in lungs, infused MSC regulate monocytes, which are extremely malleable cells and one of the first immune cell types to infiltrate into the inflamed tissue [98]. This monocyte activation would include acquisition of CD73 mRNA expression and migration to inflamed tissues in order to participate in on-site healing processes [5]. This seems to also occur when MSC are locally transplanted over injured tissues, as described by Gálvez-Montón et al. in a swine model myocardial of infarction (MI) [99]. Indeed, in this study, administered MSC attenuated inflammation and promoted myocardium healing. As mentioned above, in cell treated animals, it was later confirmed that host infiltrating monocytes de novo expressed CD73 (both control and sham animal groups lacked presence of CD73-positive monocytes) [5]. These data agree with those reported by others confirming the essential collaboration between MSC and monocytes for in vivo benefit [24,100,101,102,103,104].

## 4. Tumor Necrosis Factor (TNF)-Stimulated Protein 6 (TSG-6): Central Role in Therapeutic Benefit of Mesenchymal Stem Cells

TSG-6 is a multifunctional inflammation-associated secreted protein that can directly regulate matrix organization and structure, and it is increasingly implicated in mediating many of the immunomodulatory and tissue-protective properties of MSC (Figure 1) [105,106]. Indeed, TSG-6 expressed by MSC and recombinant TSG-6 have been shown to be efficacious in a broad range of disease indications, including mTBI [107,108]. Succinctly, first data were described in the Darwin Prockop laboratory, in 2009, when it was detected that TSG-6 secretion was upregulated in lungs by human bone marrow-derived MSC that had been injected systemically and had become trapped in the lungs in response to the inflammatory milieu present in a post-infarcted mice model [91]. The authors observed that, although administered cells arrived easily to the lungs for a targeted modulation of inflammation and were not cleared and distributed subsequently to engraft into the heart tissue, infarct size was significantly diminished in cell-treated animals. Notably, they concluded that the beneficial effect triggered by MSC was mediated by TSG-6, as they clearly demonstrated using a siRNA knockdown procedure that inhibited specifically TSG-6 expression (siTSG-6). Specifically, in mice with MI, the injection of siTSG-6 drastically abolished the mechanisms of action and associated regenerative capacity of delivered cells, whereas, in contrast, administration of an amount of recombinant human TSG-6 protein was shown to be almost as beneficial as cell treatment. Since then, additional evidence has corroborated that the secretion of TSG-6 by MSC makes a major contribution to beneficial output of MSC rather than MSC engraftment. In 2011, it was seen that bone marrow-derived MSC treated with TNF-α and other pro-inflammatory cytokines secreted high amounts of TSG-6, which attenuated the inflammatory cascade initiated by resident macrophages in a mouse peritonitis model [49]. Subsequently, Wang et al. described a TSG-6-mediated promotion of mesothelial cell repair and reduction of peritoneal adhesion in rats that underwent peritoneal scraping by intravenous injection of bone marrow-derived MSC [46]. On the contrary, the beneficial effects by MSC administration were weakened using TSG-6-RNA interfering technology. Molecularly, further progress has identified the chemokine CXCL8 (IL-8) as one of the direct ligands of TSG-6 protein that inhibited neutrophil extravasation, and thus prevented tissue damage during acute inflammation [109]. This endogenous mechanism could establish the basis for developing more efficient therapeutics to treat those inflammatory diseases in which modulation of neutrophil action is a potential target. Other studies have provided insight into the multifaceted role of MSC-released TSG-6 in wound healing. For instance, TSG-6 released from intradermally injected bone marrow MSC was shown to have anti-fibrotic effects in murine full-thickness skin wounds, which reduced myofibroblast differentiation and collagen deposition [48]. Although less than 1% of cells reached the inflamed colon of mice with colitis once injected, delivered cells formed aggregates containing macrophages and B and T cells in the peritoneal cavity, where they produced multiple immune regulatory factors, including TSG-6, which contributed to intestinal inflammation reduction [45]. Alternatively, adipose tissue MSC (ASC) also express and secrete high levels of TSG-6 in vitro and in vivo and are envisioned to improve outcomes in organ transplantation. In this regard, Kato and co-workers showed that ASC effectively attenuated damage caused by acute rejection following rat kidney transplantation by reducing T cell activation and infiltration into the allotransplanted tissue [50]. The authors clearly showed that ASC suppressed the alloresponse by means of TSG-6. Remarkably, by using a murine immune-complex vasculitis model, the therapeutic benefit of ASC has been expanded to cover tissue protection against over activated neutrophils [110]. In recent times, Wharton’s jelly and amniotic membrane have also been the subject of increasing interest for replacing bone marrow as the cell source for allogeneic transplantation, since their isolation did not require an invasive procedure for cell collection and did not raise major ethical concerns. These human tissues seem to be more primitive, proliferative, and immunosuppressive cell sources. In line with this, TSG-6 secreted by human Wharton’s Jelly MSC and amniotic membrane MSC is central to reduce the inflammatory reaction of severe burns and release of neutrophil extracellular traps associated with autoimmune and chronic inflammatory diseases, respectively [47,110]. 

Overall, these data suggest that the improvement of MSC by modulating TSG-6 expression is potentially useful for human regenerative medicine. In this context, Jha and colleagues have focused on ASC and their paracrine soluble factors as acellular therapeutic tools against mTBI. In brief, the authors first showed that concentrated conditioned media (CCM) from ASC (ASC-CCM) mitigated visual deficits and retinal inflammation in an injury model of mTBI in mice [111]. Particularly, ASC-CCM, which also exhibited high amounts of TSG-6, profoundly rescued the loss of visual acuity and contrast sensitivity four weeks post blast injury. Noticeably, this benefit seemed to be driven by the ASC-CCM anti-inflammatory properties which suppressed the activation of pro-inflammatory microglia and protected the barrier integrity of retinal endothelial cells [111]. TSG-6 knockdown ASC cultures were, then, used to generate newer CCM preparations for in vivo administration [112]. One month after intravitreal injection, the CCM derived from TSG-6 knockdown ASC did not alleviate the mTBI-mediated loss of visual properties in contrast to those injured animals treated with CCM derived from control ASC. In addition, the TSG-6-depleted ASC-CCM was not capable of reducing retinal expression of genes associated with microglial and endothelial activation, which led to vascular damage, cell death, and subsequently retinal barrier dysfunction, as properly discussed by the authors. Jha and colleagues also mechanistically confirmed the deleterious effects after abrogation of TSG-6 protein expression in ASC in in vitro transendothelial electrical resistance and pro-inflammatory signal transducer and activator of transcription 3 phosphorylation experiments. Collectively, these findings were consistent with others that demonstrated that MSC secreted high amounts of TSG-6 which, in turn, was an autocrine modulator of the therapeutic output of MSC [91,105,113,114]. For instance, Romano et al. recently described that TSG-6 paracrine function was of paramount importance for the normal activity of MSC, including proliferation rate, stemness, multipotency, and immunomodulatory capacity, and that positive isolation of MSC by means of TSG-6 expression could prompt the generation of therapeutically valuable MSC-based products [114]. On the contrary, the loss of TSG-6 protein levels could stimulate the secretion of interleukin 6 by MSC to the extracellular microenvironment enhancing adverse effects such as tumorigenesis [115]. 

## 5. Extracellular Vesicles (EV): Novel Paracrine Multifunctional Agents by Mesenchymal Stem Cells

Although we concur with Jha and colleagues on the beneficial effects of ASC-CCM on the recovery of alterations induced by mTBI, a further refining of the paracrine function of MSC can be achieved by analyzing the EV that are confined in bulk MSC secretome. As previously mentioned, EV are non-replicating, membranous vesicles secreted by the vast majority of cells, including MSC, which mimic the functions of parental cells by encapsulating soluble and membrane-bound bioactive molecules (i.e., lipids, proteins, and RNA), protected within, and thus long-term functionally active [116]. EV are specifically classified in exosomes, microvesicles, and apoptotic bodies due to their size and biogenesis. In brief, exosomes (30–200 nm) are defined as vesicles that arise from the endosomal system, by budding into the lumen of multivesicular bodies and are released when they fuse with the plasma membrane. Microvesicles, in turn, bud directly from the plasma membrane and are, then, externally secreted as larger vesicles (50 nm–1 µm). Lastly, apoptotic bodies (1–5 µm) are formed following active programmed cell death [116]. Moreover, from the point of view of their molecular signature, EV carry targeting and functional surface glycolipids and proteins, and protected cargo inside. These include proteins, metabolites, and also a variety of RNA (i.e., mRNA and microRNA), which are proposed to play key roles in many functional capacities of MSC-EV, including brain tissue repair [117,118]. Of note, small non-coding microRNA post-transcriptionally regulate the expression of hundreds of target genes and contribute to potentially alleviate central nervous system injuries [119]. For example, EV enriched with a specific microRNA cluster (miR-17-92) have contributed to neurological recovery in rats [120]. Hence, MSC-EV can resemble MSC functions, and are being considered to be a valuable alternative to MSC-based therapy. Moreover, although transfer of membrane vesicles is increasingly recognized as a pathway of intercellular communication [121,122], MSC-EV are gaining attention as possible mediators of the interaction between MSC and the cells of the immune system. In line with this, MSC-EV can be evaluated as powerful immunosuppressive agents in place of the parent MSC. Indeed, MSC-EV and MSC-CCM have been shown to differently modulate cellular response in vitro [92,123,124]. This is, for example, the case of EV-enriched preparations from Wharton’s Jelly MSC that successfully suppress T cell proliferation and do not lead to cytokine production, while their non-EV fractions induce T cell activation, with high amounts of pro-inflammatory IL-6, IFNγ, and IL-17 [92]. In light of these results, we hypothesize that the infusion of much distilled ASC-CCM could enhance the alleviation of visual abnormalities in injured animals since MSC-EV seem to be major active components of paracrine function of MSC [78,125].

Due to their size, small EV (i.e., exosomes) are gaining interest for tissue repair strategies of brain injuries such as TBI as alternatives to the MSC themselves. As mentioned previously, although hypoxia is a major hallmark of brain injuries and this could enhance the therapeutic properties of MSC, MSC-EV are non-replicative vesicles that would not change their protected molecular cargo in response to the harsh environment. Thus, in order to increase their benefit, researchers can, and aim to design and upgrade EV therapeutic cargo, targeting surface properties to also facilitate their passing across the blood-brain barrier from the circulation in the context of neurological diseases [126]. Potentially, both unmodified and engineered small MSC-EV could enter the brain parenchyma after intranasal administration and transfer their curative cargo to neuronal cells, especially given that the integrity of the blood-brain barrier is often compromised after TBI. Nevertheless, the elucidation of those surface molecules and associated signaling that regulate specific tropism and target cell discrimination by nanosized EV, remains under extensive investigation.

With that in mind, some authors have already assessed the administration of distinct MSC-EV sources in preclinical TBI models with promising results. Precisely, Zhang et al. demonstrated that the promotion of endogenous angiogenesis and neurogenesis in the hippocampus, reduced numbers of pro-inflammatory cells and improvement in cognitive and sensorimotor functional recovery by one-day post-injury intravenous administration of bone marrow MSC-EV isolated by ExoQuick-TC precipitation in rats after controlled cortical impact [127]. More recently, Williams et al. also showed that early treatment with a single dose of MSC-EV collectively attenuated brain edema, lesion size, intracranial pressure, blood-brain barrier leakage, and levels of blood-based cerebral biomarkers in a swine model of TBI and hemorrhagic shock [128]. 

In terms of EV research, parameters related to the EV isolation method, including purity and dosage, are of paramount importance for downstream EV uses. To that end, the heterogeneity and complexity of EV cargo require the design and development of high-throughput analytical platforms for accurate analysis. Presently, mass spectrometry, next-generation sequencing, and bioinformatics tools have enabled precise proteomic, transcriptomic, glycomic, lipidomic, metabolomic and genomic analyses of EV. Some studies have demonstrated how in silico system biology strategies could also be applied to further EV-mediated mechanisms of action and establish associations between genes involved in EV biogenesis/release, and human diseases and phenotypes [129]. Therefore, we strongly recommend taking advantage of size-exclusion chromatography (SEC), as an EV isolation methodology with high performance, which is easily adapted to most laboratories. SEC usually renders highly enriched EV with a highly-conserved vesicular structure, content, and function, cleaning out other potentially confusing soluble proteins or particles found in CCM (eluted in the latter SEC fractions) [130,131,132]. Importantly, this guarantees subsequent analysis by omics technologies to study their specific cargo, elucidate mechanisms of action, and predict the potential benefits or side effects upon delivery. Moreover, SEC is an appropriate method to correctly discriminate between the effects differentially induced by EV- and protein-enriched fractions from MSC [76]. Lastly, by using TSG-6 knockdown ASC cultures, Jha and colleagues could also attempt to potentially provide insight into the role of TSG-6 in the EV secreted by ASC, since a reduction in the size of EV has been found after the modulation of TSG-6 expression levels in murine bone marrow MSC [114]. This reinforces our assumption of an EV-dependent amelioration of the retinal alterations driven by the paracrine activity of ASC in the context of mTBI. In this context, it would also be of utmost interest to explore whether TSG-6 plays a key role in the regulation of EV biogenesis, cargo, and function. Whether TSG-6 is an EV-associated protein or part of the proteomic signature of distinct EV sources, including those from MSC and inflamed tissues, is under scrutiny [133]. In this sense, preliminary findings seem to validate TSG-6 to be one of the mediator factors of potent MSC-EV immunomodulation. Chaubey et al. demonstrated the presence of TSG-6 in EV preparations from umbilical cord MSC, and its contribution in recovering lung, heart, and brain alterations in a mice model of hyperoxia-induced lung injury [134]. Specifically, the authors found that EV-associated TSG-6 was responsible for the reduction of lung inflammation, alveolar-capillary leak and altered morphometry, as well as for cerebral improvements including hypomyelination reversion since beneficial effects were suppressed by siTSG-6 in the MSC or by injecting a TSG-6 neutralizing antibody along with the MSC-EV treatment. More recently, An et al. reported data supporting that TSG-6 within ASC-EV was essential for colitis symptoms alleviation by increasing the number of Tregs and macrophage polarization from M1 to M2 in the large intestine [135].

## 6. Conclusions and Future Perspectives

To conclude, although there are major challenges in regard to immune and hemocompatibility, potential safety, dosing, fitness, potency assays, and development of biomarkers predictive of response remain in the clinical use of MSC [136,137], MSC are the subject of numerous ongoing advanced trials in a wide array of human diseases. Simultaneously, researchers continue to contribute to novel advances in MSC-based cell regulation and tissue regeneration [138,139,140]. From that perspective, TBI is defined as any traumatically induced structural damage or physiological brain function disruption as a result of an external force. Following the primary injury, extensive and long-term injury is sustained through a complex cascade of secondary events. Neuroinflammation has long been considered to intensify late post-injury damage following TBI, and therefore it is proposed to be an important manageable trait in animal models and clinical trials. Although there has been dramatic progress in the management of TBI, to date there is no efficacious treatment available for patients, and morbidity and mortality remain high. Accordingly, cell therapies such as those based in MSC have been tested to increase TBI outcome. However, some limitations persist in applying MSC in the context of human regenerative therapies. These include inappropriate homing and implantation of transplanted cells, which are often entrapped in the lung barrier upon systemic administration; on-site undesired changes of the phenotype and function of infused cells; and restricted cell viability and differentiation potential. For that, many studies have moved towards the application of MSC-conditioned media, which contain a plethora of secreted soluble factors and bioactive proteins and nucleotides associated with EV that reproduce a number of effects by MSC themselves. In line with this, ASC-CCM is responsible for the recovery of the traumatic effects of blast injuries to the retina, as presented by Jha and colleagues in the context of their finely conducted studies in mice. Indeed, they expect to progress in ASC-CCM as a useful and cost-effective therapeutic solution for immediate delivery at the time of injury. Although research has to further advance in large-scale processing of MSC-EV, in obligatory good manufacturing practice conditions, quality/potency controls and instrumentation to improve visualization and quantification, the application of more purer and well-defined cell-free products such as small SEC-purified MSC-EV is also promising to boost tissue repair driven by MSC in a variety of human diseases, including mTBI, in terms of reduction of impurities, inalterability over time and usage, immunonological compatibility, and better biodistribution.

## Figures and Tables

**Figure 1 ijms-21-06761-f001:**
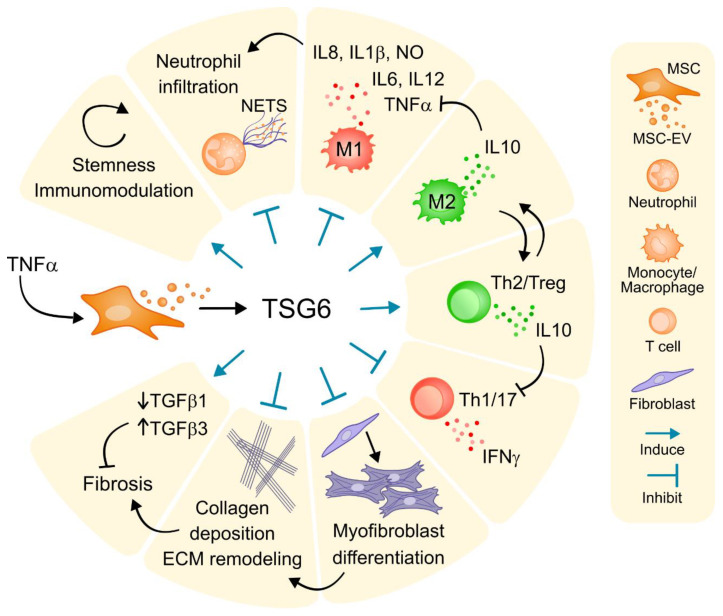
Summary of actions attributed to TSG-6, secreted by MSC and contained also in MSC-EV upon TNFα stimulation. In specific, TSG-6 can induce proliferation, stemness and increase in the immunomodulatory functions of MSC. TSG-6 reduces inflammation and switches on tissue repair by mechanisms such as: reducing neutrophil infiltration and activation; inhibiting inflammatory M1 towards M2 polarization of monocytes; inducing regulatory T cells; and reducing fibrosis. ECM: extracellular matrix; IFNγ: interferon gamma; IL: interleukin; MSC: mesenchymal stromal cell; MSC-EV: MSC-derived extracellular vesicles; NETS: neutrophil extracellular traps; NO: nitric oxide; TGFβ: transforming growth factor β; TNFα: tumor necrosis factor alpha; Treg: regulatory T cell; TSG-6: TNF-stimulated gene 6 protein.

**Table 1 ijms-21-06761-t001:** Major immune mediators by MSC.The effect on target cells is indicated as well as its expression (constitutive/inducible when specifically known, and species).

Molecule	Target	Effect	Expression	Reference
CCL2/MCP-1	Monocytes	Recruitment	Constitutive (mouse and human)	[16]
CD39, CD73	Monocytes, DC, B, T cells	Adenosine production for immune suppression	Constitutive (mouse and human)	[17,18]
CD59	MAC	Inhibition of MAC formation, final step of complement system-mediated cell lysis	Human	[19]
COX2		Production of PGE2, see PGE2		[20,21]
IL6	Monocytes	Impaired differentiation to dendritic cells	Mouse and human	[22,23]
		M2 skewing	Human	[24]
	T cells	Inhibition of mitogenic or allogeneic T cell proliferation	Mouse	[22]
	Endothelial cells	Reduced leukocyte recruitment and transendothelial migration	Human	[25]
IL10			Not expressed by murine or human MSC	[20,26,27]
		Anti-inflammatory environment, induction of Treg	Induced to monocytes (human)	[23,28,32]
	Neutrophils	Reduced neutrophil infiltrate by inducing IL10 expression to resident macrophages	Induced to resident macrophages (mouse)	[26]
HGF	Monocytes	M2 skewing	Mouse and human	[23,28]
	T cells	Inhibition of allogeneic proliferation	Constitutive (mouse)	[20]
			Constitutive (human)	[29]
HLA-G	NK cells	Reduced NK cytolytic activity	Human	[30]
	T cells	Inhibition of allogeneic proliferation and induction of Treg		
IDO	NK cells	Inhibition of IL2-induced proliferation	Human	[31]
	Monocytes	M2 skewing: increased CD206, decreased CD80; increased IL10 and decreased TNFα production	Human	[32]
	T cells	Suppression of T cell proliferation by depletion of the essential amino acid tryptophan and kynurenine accumulation	Inducible by IFNγ (mouse)	[20]
			Inducible by IFNγ (human)	[33]
Factor H	C3b	Inhibition of complement activation: blockage of C3b activation, cofactor for C3b elimination, deployment of C3 convertase (C3bBb)	Constitutive and inducible by IFNγ (human)	[34,35]
M-CSF	monocytes	M2 skewing, impaired differentiation and maturation of dendritic cells	Constitutive (human)	[24]
MMP	B cells	Reduced IgG/IgM production by MMP processing of CCL2 for reduced STAT3 and induced PAX5	Constitutive (mouse)	[36]
iNOS	T cells	Nitric oxide production for T cell suppression through inhibition of Stat5 phosphorylation	Inducible by allogeneic T cell contact (mouse)	[37]
PD-L1/2	T cells	Inhibition of proliferation and cytokine production (IL2), T cell death	Constitutive (mouse)	[38]
			Inducible by IFNγ and TNFα (human)	[20]
PGE2	NK	Inhibition of cytotoxic activity	Human	[31]
	Monocytes	M2 skewing: increased IL10 and decreased TNFα and IL6 production	Mouse	[26]
		M2 skewing	Human	[39]
		Impaired differentiation to dendritic cells and maturation		[40]
	T cells	Inhibition of allogeneic proliferation	Constitutive and increased by IFNγ and TNFα (mouse, human)	[20,41]
	T cells	Inhibition of T cell response and proliferation	Mouse *in vitro* and *in vivo* Human *in vitro*	[21,42,43]
TGFβ	T cells	Inhibition of allogeneic proliferation	Constitutive (mouse, human)	[20,29]
		Induction of Treg	Human	[24,44]
TSG6	Neutrophils	Reduced neutrophil infiltration and activation	Mouse *in vitro* and Rat *in vivo*	[45,46]
		Diminish ROS and NETs release	Human	[47]
	Monocytes	M2 skewing, limit inflammation and fibrosis	Mouse *in vitro* and *in vivo*	[48]
		Decreased NF-κB-mediated inflammatory cytokines production through CD44 receptor signaling	Human	[49]
	T cells	Suppress alloreactive T cells, attenuate acute kidney rejection	Rat *in vivo*	[50]

CCL, C-C motif chemokine ligand; CD, cluster of differentiation; COX2, cyclooxygenase 2; DC, dendritic cell; HGF, hepatocyte growth factor; HLA-G, human leukocyte antigen G; IDO, indoleamine 2,3-dioxygenase; IFN, interferon; IL, interleukin; iNOS, inducible nitric oxide synthase; MAC, membrane attack complex; MCP-1, monocyte chemoattractant protein-1; M-CSF, macrophage colony stimulating factor; MMP, matrix metalloproteinase; NETS, neutrophil extracellular traps; NFκB, nuclear factor kappa-light-chain-enhancer of activated B cells; PD-L1/2, programmed death ligand 1/2; PGE2, prostaglandin 2; ROS, reactive oxygen species; TGF, transforming growth factor; TNF, tumor necrosis factor; TSG6, tumor necrosis factor-inducible gene 6.

**Table 2 ijms-21-06761-t002:** MSC-driven effects on immune cell targets. The effects on immune cell targets of MSC are listed, with the corresponding mechanism of action in the cases it was deciphered.

Target	Mechanism	Effect	Model	Reference
Neutrophils	IL10, TSG6 SOD3	Reduction of infiltration by inducing IL10 expression to resident macrophages, blockage of CXCL8 by TSG6 production. Prevention of neutrophil death, ROS, NETs, and matrix degrading neutrophil elastase, gelatinase, and myeloperoxidase release	Mouse *in vitro* and *in vivo*	[26,45,51]
	TSG6, SOD3, HLA-G	Reduction of infiltration	Rat *in vivo*	[46]
		Decreased oxidase-1, HO-1; reduced VEGF; reduced IL8, IFNγ and increased COX2 for a dampened oxidative; vascular; and inflammatory activity. Reduced neutrophil death, ROS and NETs release	Human *in vitro*	[30,47,51]
Natural Killer (NK) cells	Contact-dep; contact-indep: IDO, PGE2, HLA-G, TGFβ1	Downregulation of NK activating receptors, inhibition of IL2-induced proliferation (through IDO), cytotoxic activity (through PGE2) and IFNγ production	Human *in vitro*	[31,41]
Monocytes/macrophages		Support survival in in vitro culture	Human *in vitro*	[24,39]
	Contact-dep; contact-indep: HGF, PGE2, TSG6	M2 skewing: increased CD206, decreased CD80; increased IL10 and decreased TNFα production	Mouse *in vitro*	[26,28,48]
	Contact-dep; contact-indep: M-CSF, HGF, PGE2		Human *in vitro*	[24,28,32,39]
	IL6	Impaired differentiation to dendritic cells and maturation: reduced CD1a, HLA-II and costimulatory molecules expression and less T cell priming	Mouse *in vitro*	[22]
	Contact-dep; contact-indep: PGE2, IL6		Human *in vitro*	[21,40]
	Partly by IL6 and M-CSF			[24,52]
Dendritic cells		Reduced CCR7 expression to inhibit migration to lymph nodes	Mouse *in vitro* and *in vivo*	[53,54]
		Reduced cross-presentation to CD8^+^ T cells	Mouse *in vivo*	[53]
		Decreased MHC-II and costimulatory molecules expression, impaired cytokine production	Mouse *in vitro* and *in vivo*	[41,53,54]
	TGF-β, IL-10, IL-6	Expression of DC costimulatory markers and ability of DCs to modulate lymphocyte proliferation	Mouse *in vitro*	[55]
T cells	NO, PGE2, IL6	Inhibition of mitogenic or allogeneic T cell proliferation	Mouse *in vitro*	[22,37,38]
	TSG6		Rat *in vitro*	[50]
			Baboon *in vitro*	[56]
	Contact-dep: PD-L1; contact-indep: PGE2, IDO, HGF, TGFβ, adenosine, HLA-G		Human *in vitro*	[18,29,30,33]
		Impaired cytotoxic activity of CD8^+^ T cells	Human *in vitro*	[44,57]
		Impaired cytotoxic activity of γΔ T cells	Mouse *in vitro*	[58]
		Upregulation of CCR7 and CD62L for retention in secondary lymphoid organs	Mouse *in vitro*	[59]
		Reduced CXCR3 (CXCL10-R) and adhesion molecules expression for reduced transendothelial migration	Human *in vitro*	[60]
	M2/MDSC induction	Shift to Th2 from Th1 or Th17 polarization	Mouse *in vitro*	[58,61]
			Human *in vitro*	[41,44]
	IDO	Induction of Tregs	Mouse *in vitro*	[62]
	Contact-dep		Human *in vitro*	[63]
	Contact-indep: TGFβ, HLA-G, PGE2	Induction of Tregs		[30,44,64]
	Need M2 skewing (CCL18 and IL10 production)			[24,39]
	IDO	Apoptosis of activated T cells	Mouse *in vitro*	[65,66,67]
		Inhibition of T cell proliferation	Human *in vitro*	[33,38,68]
		Promote survival and expansion of quiescent T cells	Mouse and human *in vitro*	[52,69,70]
B cells	Contact-dep: PD-1	Inhibition of mitogenic proliferation	Mouse and human *in vitro*	[38,71]
	IL1RA	Impaired B cell maturation and plasmablast differentiation	Mouse and human *in vitro*	[71,72]
	MMP processing of CCL2 for reduced STAT3 activation and induced PAX5 transcription	Reduced production of IgG and IgM under strong stimulation	Mouse *in vitro*	[36]
			Human *in vitro*	[73,74]
	Contact-dep; contact-indep: IDO	Induction of Bregs	Mouse and human *in vitro*	[71,75,76,77,78]

Abbreviations meaning as they appear. Breg, regulatory B cell; CCR7, C-C motif chemokine receptor 7; CD, cluster of differentiation; CXCL, C-X-C motif chemokine ligand; IL, interleukin; HGF, hepatocyte growth factor; HLA, human leukocyte antigen; HO-1, heme oxygenase-1; IDO, indoleamine 2,3-dioxygenase; IFN, interferon; COX2, cyclooxygenase-2; M-CSF, macrophage colony stimulating factor; MHC, major histocompatibility complex; MDSC, myeloid-derived suppressor cell; NETS, neutrophil extracellular traps; NO, nitric oxide; PAX5, paired box protein 5; PGE2, prostaglandin E2; Treg, regulatory T cell; PD-1, programmed death-1; ROS, reactive oxygen species; SOD3, superoxide dismutase; STAT3, signal transducer and activator of transcription 3; TGF, transforming growth factor; TNF, tumor growth factor; TSG6, tumor necrosis factor-inducible gene 6; VEGF, vascular endothelial growth factor.

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
