# Peer review of "Potential of Extracellular Vesicle-Associated TSG-6 from Adipose Mesenchymal Stromal Cells in Traumatic Brain Injury"

_ijms, 2020, doi:10.3390/ijms21186761_

Round 1
Reviewer 1 Report
The Manuscript “Distilling the paracrine function of mesenchymal stem cells to boost therapeutic potential” nicely describes immunomodulatory functions of MSC, particularly in traumatic brain injury. The paper is correctly written, but there are several flaws:
Major points:
It would be nice that Authors extend their description of hypoxia/anoxia conditions upon TBI, since these factors can strongly impact MSC functions.
Please, correct the abbreviation for MSC. This term should be used for mesenchymal stromal cell, rather than mesenchymal cells. This is mandatory, since in the majority of cited reference, no stem cell features evidenced for MSC.
Author Response
Comments to the Author
The Manuscript “Distilling the paracrine function of mesenchymal stem cells to boost therapeutic potential” nicely describes immunomodulatory functions of MSC, particularly in traumatic brain injury. The paper is correctly written, but there are several flaws:
Major points:
- It would be nice that Authors extend their description of hypoxia/anoxia conditions upon TBI, since these factors can strongly impact MSC functions.
Answer: We thank reviewer for appreciating our work. Following reviewer’s suggestion, this relevant concern raised by reviewer is now highlighted:
Page 3, lines 102-106: “In this context, cerebral hypoxia, understood as a decrease of oxygen supply to the brain, can develop as a secondary complication over a period of hours to days after TBI, being more frequent in moderate or severe injuries. Remarkably, these hypoxia episodes can aggravate initial brain damage (with increased neuronal death and neuroinflammation), and correlate with worse neurological outcomes…”
Page 3, lines 128-135: “It is well established that oxygen deprivation, creating a hypoxic environment, is crucial to influence the properties of MSCs [15]. This is, for example, the case of the enhanced MSC’s proliferation and angiogenic potential observed in hypoxia pre-conditioned MSC. Also, under hypoxic conditions, MSC are induced to secrete more lactate, which polarises macrophages to an anti-inflammatory phenotype, and exhibit great amounts of IFNg-mediated IDO and their capacity to inhibit T cell activation. Thus, in the context of TBI, the absence of oxygen could be responsible, in part, for the therapeutic benefit of MSC therapy, including their immunomodulatory capacity…”
Page 11, lines 315-323: “…Due to their size, small EV (i.e. exosomes) are gaining interest for recovery-enhancing treatment of brain injuries such as TBI as alternative of the MSC themselves. As mentioned previously, although the absence of oxygen is a major hallmark upon brain injuries and this could greatly enhance outcomes after MSC therapy, derived EV are non-replicative membrane-contained vesicles that theoretically protect their molecular cargo from the activity of environmental factors and it is thus assumed that EV content cannot be altered once secreted. However, in order to increase benefit, researchers aim to alter inner EV content or surface properties to facilitate their passing across the blood-brain barrier from the circulation in the context of neurological diseases [126]. Potentially, unmodified or engineered small EV could enter the brain parenchyma after intranasal administration and transfer their curative cargo to neuronal cells because integrity of blood-brain barrier is often compromised directly by TBI. Nevertheless, the elucidation of those surface molecules and associated signalling regulating specific tropism and target cell discrimination by nanosized EV remains under extensive investigation.”
- Please, correct the abbreviation for MSC. This term should be used for mesenchymal stromal cell, rather than mesenchymal cells. This is mandatory, since in the majority of cited reference, no stem cell features evidenced for MSC.
Answer: We are grateful for reviewer’s comment. As reviewer suggested, the term MSC now refers to as those fibroblast-like, plastic-adherent, multipotent mesenchymal stromal cells (regardless of the tissue from which they are isolated) that do often unmeet stem cell criteria, according to The International Society for Cellular Therapy recommendation on the clarification of the nomenclature for MSC (Horwitz et al. Cytotherapy 2005).
Thus, this revised version of the manuscript has been re-written to remove the term ‘mesenchymal stem cell’ and to read as follows:
Page 2, lines 55-58: “Mesenchymal stromal cells (MSC) constitute a heterogeneous cell population of non-hematopoietic progenitors, with a fibroblast-like appearance, plastic adherence capacity, ability to differentiate into tissues of mesodermal lineages and a well-established cell surface expression pattern, regardless of the tissue from which they are collected…”
Reviewer 2 Report
The work of Santiago Roura et al. reviews the therapeutic potential of MSC-derived paracrine signals for the treatment especially of deficits after mild traumatic brain injury. Furthermore, the authors focus on the current knowledge about gene-6 (TSG-6) stimulated by tumor necrosis factor (TNF) and the extracellular vesicle (EV) -based mechanisms that govern the immunosuppressive and regenerative activity of long duration of MSCs.
The work is of interest to Journal readers because it highlights the growing therapeutic interest in products secreted by MSCs as anti-inflammatory and regenerative agents. However, in my way of understanding, there are some aspects that should be modified:
-The title of the work is too generic. The authors should make more specific mention of the subject: mild traumatic brain injury, ASC, stimulated gene-6 (TNF) and extracellular vesicles (EV), etc.
-I think it would be convenient to unify Tables 1 and 2, because readers could be confused with the concepts they are intended to describe. Furthermore, the information could refer only to human MSCs, discarding those of natural origin.
-In contrast, a new Table could be elaborated for the actions attributable to TSG-6, so that the reading of chapter 4. “TSG-6: central role in therapeutic benefit of mesenchymal stem cells”, is easier.
- The authors could define more fully the types of EV, highlighting the exosomes, which have the interest of crossing the blood-brain barrier, as well as mention the content of these particles, especially the micro-RNAs as the main responsible for their capacity functional.
-Although the review is focused on the ASC-CCM, the authors could mention the interest of MSC from other origins.
-The connection between TSG-6 and EV should be more convincingly justified.
Author Response
Comments to the Author
The work of Santiago Roura et al. reviews the therapeutic potential of MSC-derived paracrine signals for the treatment especially of deficits after mild traumatic brain injury. Furthermore, the authors focus on the current knowledge about gene-6 (TSG-6) stimulated by tumor necrosis factor (TNF) and the extracellular vesicle (EV) -based mechanisms that govern the immunosuppressive and regenerative activity of long duration of MSCs.
The work is of interest to Journal readers because it highlights the growing therapeutic interest in products secreted by MSCs as anti-inflammatory and regenerative agents. However, in my way of understanding, there are some aspects that should be modified:
- The title of the work is too generic. The authors should make more specific mention of the subject: mild traumatic brain injury, ASC, stimulated gene-6 (TNF) and extracellular vesicles (EV), etc.
Answer: We thank reviewer for appreciating our work. Done.
- I think it would be convenient to unify Tables 1 and 2, because readers could be confused with the concepts they are intended to describe. Furthermore, the information could refer only to human MSCs, discarding those of natural origin.
Answer: We are grateful for reviewer’s suggestion. We respectfully assume that, with this approach, readers focused or interested differently in either molecular or cellular immunology by means of MSC or both can go separately or not throughout tables as they prefer to find, in more detail or in depth, those aspects more specifically related to their respective areas.
- In contrast, a new Table could be elaborated for the actions attributable to TSG-6, so that the reading of chapter 4. “TSG-6: central role in therapeutic benefit of mesenchymal stem cells”, is easier.
Answer: Following reviewer’s concern, the potential mechanisms by means of EV-associated TSG-6 from MSC in immune modulation and tissue repair are alternatively depicted in the new schematic image (see Figure 1) accompanying chapter 4 in this improved version of the manuscript. This approach could be even more valuable for potential readers.
- The authors could define more fully the types of EV, highlighting the exosomes, which have the interest of crossing the blood-brain barrier, as well as mention the content of these particles, especially the micro-RNAs as the main responsible for their capacity functional.
Answer: As suggested by reviewer, this improved version of the manuscript has been re-written as follows:
Page 11, lines 315-323: “…EV are specifically classified in exosomes, microvesicles and apoptotic bodies due to their size and biogenesis. In brief, exosomes (30–200 nm) are defined as vesicles that arise from the endosomal system, by budding into the lumen of multivesicular bodies and released when these fuse with the plasma membrane. Microvesicles, in turn, bud directly from the plasma membrane and are then externally secreted as larger vesicles (50 nm-1 µm). Lastly, apoptotic bodies (1-5 µm) are formed following active programmed cell death [116]…”
Page 11, lines 324-331: “…Moreover, from the point of view of their molecular signature, EV carry targeting and/or functional surface glycolipids and proteins, and protected cargo inside. These include proteins, metabolites and also a variety of RNA (i.e. mRNA and microRNA), which are proposed to play key roles in many functional capacities of MSC-EV, including brain tissue repair [117,118]. Of note, small non-coding microRNA post-transcriptionally regulate the expression of hundreds of target genes and contribute to potentially alleviate central nervous system injuries [119]. For example, EV enriched with a specific microRNA cluster (miR-17-92) contribute to neurological recovery in rats [120]…”
Page 11, lines 344-359: “…Due to their size, small EV (i.e. exosomes) are gaining interest for tissue repair strategies of brain injuries such as TBI as alternative of the MSC themselves. As mentioned previously, although hypoxia is a major hallmark of brain injuries and this could enhance the therapeutic properties of MSC, MSC-EV are non-replicative vesicles that would not change their protected molecular cargo in response to the harsh environment. Thus, in order to increase their benefit, researchers can, and aim to design and upgrade EV therapeutic cargo, targeting surface properties to also facilitate their passing across the blood-brain barrier from the circulation in the context of neurological diseases [126]. Potentially, both unmodified and engineered small MSC-EV could enter the brain parenchyma after intranasal administration and transfer their curative cargo to neuronal cells, especially given the integrity of blood-brain barrier is often compromised after TBI. Nevertheless, the elucidation of those surface molecules and associated signalling regulating specific tropism and target cell discrimination by nanosized EV remains under extensive investigation”
- Although the review is focused on the ASC-CCM, the authors could mention the interest of MSC from other origins.
Answer: We are grateful for the reviewer’s suggestion. Following reviewer’s recommendation, the original version of manuscript has been modified to include a number of studies reporting data related to the topic using other MSC-EV sources.
- The connection between TSG-6 and EV should be more convincingly justified.
Answer: We are also grateful for the reviewer’s comment. Following reviewer’s recommendation, this improved version of the manuscript reads as follows:
Page 12, lines 389-402: “…In this context, it would also be of utmost interest to explore whether TSG-6 plays a key role in the regulation of EV biogenesis, cargo and/or function. Whether TSG-6 is an EV-associated protein or part of the proteomic signature of distinct EV sources, including those from MSC and inflamed tissues, is under scrutiny [133]. In this sense, preliminary findings seem to validate TSG-6 as a one of the mediator factors of potent MSC-EV immunomodulation. Chaubey et al. have demonstrated the presence of TSG-6 in EV preparations from umbilical cord MSC, and its contribution in recovering lung, heart and brain alterations in a mice model of hyperoxia-induced lung injury [134]. In specific, the authors found that EV-associated TSG-6 was responsible for the reduction of lung inflammation, alveolar-capillary leak and altered morphometry, as well as for cerebral improvements including hypomyelination reversion since beneficial effects were suppressed by siTSG-6 in the MSC or by injecting a TSG-6 neutralizing antibody along with the MSC-EV treatment. More recently, An et al. reported data supporting that TSG-6 within ASC-EV is essential for colitis symptoms alleviation by increasing the number of Tregs and macrophage polarisation from M1 to M2 in the large intestine [135].”
Reviewer 3 Report
Dear Authors,
In your manuscript entitled “Distilling the paracrine function of 2 mesenchymal stem cells to boost 3 therapeutic potential” you extend the discussion by Jha and colleagues on the beneficial effects of MSCs on immunomodulation and tissue repair. In particular, on the basis of your research experiences, you discuss on tumor necrosis factor (TNF)-stimulated gene-6 (TSG-6) and extracellular vesicles (EV)-based mechanisms governing MSC’ long-lasting immunosuppressive and regenerative activity. The idea and the project are very interesting considering primarily the possible role of extracellular vesicles (EV) and the mechanisms of immunomodulation involving the MSCs. Despite this, the manuscript is however not well organized starting the title.
Major revisions:
The title should rewritten on the basis of the contents of the entire manuscript to be more specific (referring in particular to the potential therapeutic use of EV-MSCs in TBI). Moreover, it is required to give more personality to the article.
Moreover, the different chapters should be more combined and interlinked. The chapter relative to TSG-6 should require an image to explain better the roles and the benefits for MSCs.
It should needed to introduce an image explaining the potential role of EV-MSCs in the therapy of TBI.
Minor revisions:
In the introduction the authors should cite other the paracrine effect of MSCs (page 2 line 57) also the potential autocrine effects that have repercussions on conditioned medium and on MSC “behavior” (Pelagalli A. et al., Autocrine signals increase ovine mesenchymal stem cells migration through Aquaporin-1 and CXCR4 overexpression. J Cell Physiol. 2018 Aug;233(8):6241-6249.)
In addition, some sentences in different chapters are not clear and require to be rewritten (see page 2 lines 82-85) (page 8 lines 176-179)
The manuscript requires major revisions.
Author Response
Comments to the Author
In your manuscript entitled “Distilling the paracrine function of 2 mesenchymal stem cells to boost therapeutic potential” you extend the discussion by Jha and colleagues on the beneficial effects of MSCs on immunomodulation and tissue repair. In particular, on the basis of your research experiences, you discuss on tumor necrosis factor (TNF)-stimulated gene-6 (TSG-6) and extracellular vesicles (EV)-based mechanisms governing MSC’ long-lasting immunosuppressive and regenerative activity. The idea and the project are very interesting considering primarily the possible role of extracellular vesicles (EV) and the mechanisms of immunomodulation involving the MSCs. Despite this, the manuscript is however not well organized starting the title.
- The title should be rewritten on the basis of the contents of the entire manuscript to be more specific (referring in particular to the potential therapeutic use of EV-MSCs in TBI). Moreover, it is required to give more personality to the article.
Answer: We are grateful for reviewer’s recommendation. Done.
- Moreover, the different chapters should be more combined and interlinked. The chapter relative to TSG-6 should require an image to explain better the roles and the benefits for MSCs.
Answer: We are also grateful for the reviewer’s suggestion. Accordingly, this improved version of the manuscript includes a new schematic figure summarizing the potential TSG-6-mediated immunomodulatory and tissue repair properties of MSC and derived EV (see Figure 1).
- It should need to introduce an image explaining the potential role of EV-MSCs in the therapy of TBI.
Answer: Done. Overall immunomodulatory and tissue repair effects by means of MSC-secreted TSG-6, either as soluble factor or contained in their derived EV, which are depicted in Figure 1 are assumed to be also central in the context of therapy in TBI.
- In the introduction the authors should cite other the paracrine effect of MSCs (page 2 line 57) also the potential autocrine effects that have repercussions on conditioned medium and on MSC “behavior” (Pelagalli A. et al., Autocrine signals increase ovine mesenchymal stem cells migration through Aquaporin-1 and CXCR4 overexpression. J Cell Physiol. 2018 Aug;233(8):6241-6249.)
Answer: Done.
- In addition, some sentences in different chapters are not clear and require to be rewritten (see page 2 lines 82-85) (page 8 lines 176-179)
Answer: Done.
Round 2
Reviewer 2 Report
The authors answered all the questions
Reviewer 3 Report
Dear authors, the revised version of the manuscript was qualitatively improved according the reviewers suggestions. Many points were largely argumented and many points were discussed in more appropriate manner.